# Development of Fixed-Wing UAV 3D Coverage Paths for Urban Air Quality Profiling

**DOI:** 10.3390/s22103630

**Published:** 2022-05-10

**Authors:** Qianyu Zhou, Li-Yu Lo, Bailun Jiang, Ching-Wei Chang, Chih-Yung Wen, Chih-Keng Chen, Weifeng Zhou

**Affiliations:** 1Department of Aeronautical and Aviation Engineering, The Hong Kong Polytechnic University, Kowloon, Hong Kong; q1zhou@polyu.edu.hk (Q.Z.); liyu.lo@connect.polyu.hk (L.-Y.L.); bailun-robin.jiang@connect.polyu.hk (B.J.); chihyung.wen@polyu.edu.hk (C.-Y.W.); 2Department of Mechanical Engineering, The Hong Kong Polytechnic University, Kowloon, Hong Kong; chingwei.chang@connect.polyu.hk; 3Department of Vehicle Engineering, National Taipei University of Technology, Taipei 10608, Taiwan; ckchen@ntut.edu.tw; 4School of Professional Education and Executive Development, The Hong Kong Polytechnic University, Kowloon, Hong Kong

**Keywords:** air quality, monitoring, fixed-wing UAV, coverage path planning

## Abstract

Due to the ever-increasing industrial activity, humans and the environment suffer from deteriorating air quality, making the long-term monitoring of air particle indicators essential. The advances in unmanned aerial vehicles (UAVs) offer the potential to utilize UAVs for various forms of monitoring, of which air quality data acquisition is one. Nevertheless, most current UAV-based air monitoring suffers from a low payload, short endurance, and limited range, as they are primarily dependent on rotary aerial vehicles. In contrast, a fixed-wing UAV may be a better alternative. Additionally, one of the most critical modules for 3D profiling of a UAV system is path planning, as it directly impacts the final results of the spatial coverage and temporal efficiency. Therefore, this work focused on developing 3D coverage path planning based upon current commercial ground control software, where the method mainly depends on the Boustrophedon and Dubins paths. Furthermore, a user interface was also designed for easy accessibility, which provides a generalized tool module that links up the proposed algorithm, the ground control software, and the flight controller. Simulations were conducted to assess the proposed methods. The result showed that the proposed methods outperformed the existing coverage paths generated by ground control software, as it showed a better coverage rate with a sampling density of 50 m.

## 1. Introduction

Over the past few decades, rapid economic growth has resulted in skyrocketed population growth and industrialization in many major metropolitan areas. The increase in urbanization has led to a series of environmental consequences, with the drastic increase in atmospheric pollutants being one of them. Undoubtedly, the deterioration in air quality has become a global issue and has caused adverse impacts on both the environment and society, demanding immediate resolutions. Li et al. [1] further advocated that the global air pollution issue requires different means to resolve depending on the location since the causes vary; for instance, the PM 2.5 particles in Asian countries are mainly caused by heating and cooking, whereas agriculture is the culprit in European countries. This hence makes monitoring the pollutant concentration and distribution essential so that the composition of toxicities can be further quantified [2].

Currently, environmental authorities around the globe have established methods to measure and monitor atmospheric indicators. For instance, the Environment Protection Department (EPD) of the Hong Kong Special Administrative Region (HKSAR) has been tracking street-level pollution as well as the regional smog problem through the installation of ground monitoring stations and publishing the measured data [3]. The method has generally been applicable with high robustness and could be seen in various places; nevertheless, such a conventional method is deemed spatio-temporally inflexible, as the sampling points are stationary and limited, while skilled personnel are required to conduct measurements with higher accuracy [4]. Therefore, many research and governmental institutions further employ crewed aircraft or satellites for larger-scale monitoring activities [4], such as the work proposed or mentioned by [5,6,7,8]. Although it is considered that satellites and aerial sensors can cover more sampling points and provide complete profiling information, the higher deployment and maintenance cost of those technologies also brings inaccessibility to the general public, making the task operationally difficult [4].

In recent years, with the advances in unmanned aerial vehicles (UAVs), many researchers have proposed to utilize the technology to carry out monitoring jobs, as they have lower cost, higher autonomy, and easier availability [9]. Gu et al. [10] developed an end-to-end quadrotor-based air pollution profiling system. They tried to reach full autonomy by integrating the drone, the ground station software, the sensors, and the data acquisition and fusion modules. Similarly, the authors in [11] attempted to track and analyze the SO_2_ and NO_2_ of ship plumes by exploiting a rotary unmanned aerial vehicle and embedded air quality sensors. Based on a similar configuration, Araujo et al. [12] programmed different flight patterns for a commercial drone to evaluate the efficacy of measuring air pollutants. Jumaah et al. [13] also developed a UAV-based air monitoring system, where a hex-rotor was utilized to assess PM2.5 particles. Similar work applying UAVs for air quality surveying can also be found in [14,15,16,17,18]. Furthermore, among the aforementioned projects, many adopt coverage path planning (CPP) methods to ensure a high coverage rate of a certain region of interest (ROI). The method can be frequently seen in several robotic applications, such as vacuum robots [19], photogrammetry drones [20], or automatic lawnmowers [21], in which paths are searched and determined to visit all interest points within the ROI [22]. In specific, to acquire such a path, CPP can be modeled as a traveling salesman problem (TSP) [22], in which the NP-hard problem is then solved with heuristic or complete methods [23]; the final output then allows the agent to travel all points on the network without duplicated visits. For a deeper understanding of the traveling salesman problem, we refer readers to [23,24].

Despite a considerable body of research on air quality monitoring rotary drones utilizing coverage path planning methods, we argue that the practicality of such a system design is still discounted. Particularly, due to the size, weight, and power (SWaP) limitations of most currently commercialized drones, the payload, range, and endurance are usually restricted [25], making large-scale air quality monitoring missions temporally and spatially unsustainable [4]. In contrast, fixed-wing-based UAVs provide a plausible alternative; in recent years, several research studies on fixed-wing aircraft for remote sensing applications have been published. For instance, Simon et al. [26] utilized a tail-sitter configuration UAV to conduct 3D mapping of the terrain of a specific region, in which they tried to directly acquire photogrammetric information via a lightweight VTOL aerial vehicle. Work conducted by Coombes et al. [27,28,29,30], on the other hand, developed coverage path planning methods through geometry decomposition, Boustrophedon paths, etc., and meanwhile, different scenarios were addressed, including utilization in agriculture activities and windy situations. Paull et al. [31] also proposed a path planner that attempted to achieve a coverage path without a priori understanding of the workspace, in which a camera sensor was equipped, and an in situ path planning module was included for the aircraft to make control decisions. In addition to all of the above, Yu et al. [32] utilized the knee-guided differential evolution algorithm [33] to construct a path planning problem for UAVs for disaster scenarios within 3D terrain situations. In particular, the work aimed to solve the problem by utilizing B-spline paths and modeling the task with multi-objective optimization, where distance and risk are mainly considered as the objective functions; such a method could then offer an optimal solution and allow the vehicle to follow a smooth path.

Therefore, motivated by the pioneering works, this study utilized a vertical takeoff and landing (VTOL) configuration UAV to conduct a 3D air quality profiling task, as it is deemed to have a higher range and longer endurance than rotary UAV, as well as a lower cost and easier deployment (requires no open area or takeoff runway) than crewed aircraft. Particularly, this research aimed to develop an easy-access coverage path planning module, which automatically generates a kinematic feasible path for a VTOL UAV of quad-plane configuration for 3D air quality profiling. In addition, to make the proposed system more accessible to the public, a user interface was added for all users from different fields to deploy the VTOL UAV. Therefore, the objectives were as follows:To design a 3D coverage path planning algorithm that outputs a path aligning with the quad-plane physical feasibilities based upon a 3D voxel region of interest (ROI) in GPS coordinates, which are inputted according to user specifications;In addition to the planner, design an easy-access user interface written in python script for users to interact with the module;To carry out simulations within the software-in-the-loop platform, in which the efficiency of our method is assessed by comparing it with the default software paths.

To address the above, the remainder of this article is organized as follows. Section 2 outlines the significant materials and methodologies of the coverage path planning for 3D profiling. Then, Section 3 discusses the simulation results. Lastly, Section 4 projects possible future work and improvements.

## 2. Materials and Methods

In order to perform a 3D air quality profiling of an ROI, the proposed work consists of the following modules: (1) coverage path generation for 3D air quality profiling, (2) secure return to launch path generation, (3) an easy-access user interface, and (4) ground control station software.

### 2.1. Coverage Path Generation for 3D Air Quality Profiling

As mentioned in Section 1, currently, there is no optimal coverage path planning method for fixed-wing UAVs; most flight missions are either generated manually by human input or by the default photogrammetric purpose auto-grid method from ground control software. Currently, ArduPilot Mission Planner [34] and QGroundControl [35] are two of the most widespread open-source software. These methods, when being applied to fixed-wing aircraft, have disadvantages of the following: (1) low efficiency, as the waypoints are required to be manually inputted through ground control software [34,35]; (2) empiricism dependency, where operators should be equipped with basic knowledge of coverage paths so that the ROI can be completely surveyed; and (3) lack of compliance with the dynamic feasibilities of fixed-wing vehicles, as in photogrammetric purpose auto-grid paths [20], the methodology is mainly designed for rotary UAVs, and applicability on fixed-wing planes can be suboptimal. Hence, this study asserts the importance of an automatic calculation- and theory-based coverage path planning algorithm, in particular for air quality sensing missions. The following further elucidates the methodologies of the proposed coverage path planner.

#### 2.1.1. Coverage Path Planning Constraints for Fixed-Wing Aircrafts

To perform 3D ROI air quality profiling, the coverage path should be designed to cover all the voxels or sampling points within an ROI as densely as possible, i.e., maximizing the coverage rate. For the common coverage path planning of an ROI in quadrotor applications, many adopt the Boustrophedon path [36,37], in which the vehicle follows an iterating back and forth path to cover the ROI, as shown in Figure 1.

In this study, the concept of the Boustrophedon path was utilized as well so that a selected ROI can be covered sufficiently. However, different from rotary aerial vehicles, due to their dynamic system, fixed-wing planes have restricted agility; hence, it makes the aforementioned method only partially applicable in this scenario. Amongst the limitations, the turning radius is a significant factor. The turning radius of a fixed-wing plane can be derived as:
(1)
$$F_{horizontal}=T\sin\beta =ma=m\frac{v^{2}}{R}=m\omega^{2}R$$


(2)
$$F_{vertical}=mg=T\cos\beta .$$


In which, the vertical force is assumed to be the weight of the aircraft. By dividing Equation (1) by Equation (2), we obtain:
(3)
$$\frac{F_{horizontal}}{F_{vertical}}=\frac{T\sin\beta }{T\cos\beta }=\frac{m\frac{v^{2}}{R}}{mg}=\frac{m\omega^{2}R}{mg}=\tan\beta .$$


Thus,

(4)
$$R=\frac{v^{2}}{g\tan\beta }.$$


As shown above in Figure 2, 
T
 represents the lift, 
m
 is the mass of the aircraft, 
v
 stands for the tangent velocity during turning, 
ω
 denotes the angular velocity, 
R
 indicates the turning radius, and lastly, 
β
 is the banking angle. Equation (4) then shows the minimum turning radius of a fixed-wing plane; in particular, the tangent value of the banking angle 
β
 is equivalent to the ratio of the horizontal and vertical force, indicating a limitation on the turning radius 
R
. This results in low robustness in tracking the Boustrophedon paths. To resolve the low maneuverability, the Dubins path [38,39] was introduced. The Dubins path is an optimal path where the curve is in compliance with the curvature constraints at the initial and final waypoints, and it is proven to provide the shortest path for a forward traveling vehicle or robot [38,39]. Therefore, based on the Boustrophedon path, the Dubins path was added during the turning stage, so that the aircraft could have a higher tracking performance at all stages. Moreover, for the Dubins path, there exists an array of path types that are represented by 
L
, 
S
, and 
R
, indicating ‘left turn’, ‘straight’, and ‘right turn’, respectively; the turning paths are usually denoted as 
$D=\left\{LSL,\;\;RSR,\;\;RSL,\;\;LSR,\;\;RLR,\;\;and\;LRL\right\}.$
 A visual representation is shown below in Figure 3.

Based on the Boustrophedon and Dubins paths, the following sections introduce the proposed coverage path for 3D air quality profiling.

#### 2.1.2. Cycle-Boustrophedon Path Planning

Given a specified cubic ROI with vertices 
$v=\left\{v_{1},v_{2},v_{3},v_{4}\right\}$
 in GPS coordinates and altitude boundaries of 
$h_{max}$
 and 
$h_{min}$
 in meters, a coverage path was planned to complete an air quality profiling task with a predefined sampling density 
d
 (all notations 
d
 in this manuscript stand for the sampling density as described here). To perform the calculation, coordination transformation (illustrated in Figure 4) was first conducted:
(5)
$$v_{\theta }=\left[\begin{array}{cc} \cos\theta & -\sin\theta \\ \sin\theta & \cos\theta \end{array}\right]v\;\;$$

where 
θ
 is calculated via the relative rotation angle of the ROI to the global frame. The path was then planned within the local frame and rotated back afterward.

For a conventional Boustrophedon path, each parallel track is executed in series adjacently; however, in this scenario, to have a smoother turn for fixed-wing planes, we first segregated the tracks into two sub-Boustrophedon groups, even and odd orders, where they would be followed consecutively. Consequently, each level of altitude was required to complete two ‘cycles’ of paths, hence the name ‘cycle-Boustrophedon’. Figure 5a shows the original Boustrophedon method, whilst the illustration on the right (Figure 5b) displays the proposed method. Specifically, the blue, which is the ‘even order’, is first implemented, and then the paths in orange, which are the ‘odd order’, are followed. Notably, the “odd” and “even” orders refer to the horizontal path orders from top to bottom in Figure 5b. Furthermore, to fulfill the profiling density, the distance between the abutting tracks should be 
d
, making the distance between the tracks in the sub-Boustrophedon group 
2d
. The proposed planning was duplicated vertically in layers, with the number of layers being n = 
$\frac{h_{max}-h_{min}}{d}+1$
. We believed that this addition to the conventional Boustrophedon paths would allow the fixed-wing aircraft to have a more optimal performance, since the majority of the sampling density possesses a smaller scale than the minimum radius of the UAV.

After setting the main tracks of the proposed cycle-Boustrophedon paths, Dubins paths were then added at the turning points for further path optimization. As there exists a minimum turning radius (which is defined as 
$r_{1}$
 below) for a fixed-wing plane, and since for each turning point, the aircraft is required to make a 180-degree turn, an LRL/RLR or LSL/RSR type Dubins path was added to the sub-Boustrophedon path. The determination of using either the LRL/RLR or LSL/RSR type Dubins path depended on the user predefined sampling density, as:
(6)
$$\left\{\begin{array}{c} d<R,\;\;D_{dubins}\in LRL/RLR\\ d\geq R,\;D_{dubins}\in LSL/RSR \end{array}\right.$$


Figure 6a,b above illustrate the two aforementioned types. To perform the above, waypoints were further defined. For the LRL/RLR with the RLR, for instance as shown in Figure 7, for each turning, the following points (from a bird’s eye view) were first defined:
(7)
$$assume\;point\;A=\left\{a_{1},\;a_{2}\right\},\;\&\;\alpha =\arccos\left(\frac{r_{1}+d}{2r_{1}}\right),\;r_{1}=R=\frac{V^{2}}{g\tan\beta }.$$


Hence,

(8)
$$\left\{\begin{array}{c} A=\left(a_{1},a_{2}\right)\\ B=\left(a_{1}+r_{1}\;sin\alpha ,\;a_{2}+r_{1}\;\left(\cos\alpha -1\right)\right)\\ C=\left(a_{1}+r_{1}\left(2\sin\alpha +1\right),a_{2}+d\right)\\ B^{\prime }=\left(a_{1}+r_{1}\left(\sin\alpha \right),a_{2}+r_{1}\left(1-\cos\alpha \right)+2d\right)\\ A^{\prime }=\left(a_{1},\;a_{2}+2d\right) \end{array}.\right.$$


The waypoint radius was further considered to obtain better performance, apart from the above-listed waypoints. The waypoint radius is a default parameter for flight controllers to justify whether the system has reached the desired waypoint; for most fixed-wing aircraft, as they have lower maneuverability, the waypoint radius is usually set at a large scale (compared to rotary vehicles). Therefore, in this scenario, where each waypoint is relatively close to another, the waypoint radius 
$r_{2}$
 was further considered when conducting path planning. Figure 8 shows the final waypoints.

The coordinates are:
(9)
$$\left\{\begin{array}{c} A=\left(a_{1},a_{2}\right)\\ B_{1}=\left(a_{1}+r_{1}\;sin\alpha ,\;a_{2}+r_{1}\;\left(\cos\alpha -1\right)-r_{2}\right)\\ C_{1}=\left(a_{1}+r_{1}\left(2\sin\alpha +1\right)+r_{2},a_{2}+d\right)\\ B_{1}^{\text{’}}=\left(a_{1}+r_{1}\left(\sin\alpha \right),a_{2}+r_{1}\left(1-\cos\alpha \right)+2d+r_{2}\right)\\ A^{\prime }=\left(a_{1},\;a_{2}+2d\right) \end{array}\right..$$


Therefore, to complete one iteration of the cycle-Boustrophedon path, there were both RLR and LRL Dubins paths and 10 waypoints in total, as shown in Figure 9 (*A*→*B*→*C*→*B’*→*A’*→*D*→*E*→*F*→*E’*→*D’*).

For the LSL/RSR, as shown in Figure 6b, taking LSL for instance, fewer waypoints were needed; these waypoints are defined as:
(10)
$$\left\{\begin{array}{c} A=\left(a_{1},a_{2}\right)\\ B=\left(a_{1}+r,\;a_{2}+r\right)\\ B^{\prime }=\left(a_{1}+r,\;a_{2}+2d-r\right)\\ A^{\prime }=\left(a_{1},\;a_{2}+2d\right) \end{array}.\right.$$


Similarly, as the waypoints were relatively close to each other, the waypoint radius should also be considered, making the final waypoints:
(11)
$$\left\{\begin{array}{c} A=\left(a_{1},a_{2}\right)\\ B_{1}=\left(a_{1}+r_{1}+r_{2},\;a_{2}+r_{1}\right)\\ B_{1}^{’}=\left(a_{1}+r_{1}+r_{2},\;a_{2}+2d-r_{1}\right)\\ A^{\prime }=\left(a_{1},\;a_{2}+2d\right) \end{array}\right..$$


Therefore, for each iteration, there were both LSL and RSR Dubins paths and 8 waypoints in total in the proposed method (shown in Figure 10, *A*→*B*→*B’*→*A’*→*C*→*D*→*D’*→*C’*).

The following pseudo-code shows the overall cycle-Boustrophedon path planning Algorithm 1:
**Algorithm 1** Cycle-Boustrophedon Path Planning.
**Input:**
 Home and takeoff location
 ROI vertices latitude and longitude
 Path separation distance in meters: 
s

 ROI minimum altitude in meters: 
hmin

 Number of layers: 
nh

 Layer separation distance in meters: 
hs

**Output:** Readable waypoint file for GCS software

 Transfer vertices to local coordinates.
 Calculate LongEdge and ShortEdge
 
$n=ceil\left(\frac{ShortEdge}{4s}\right)$
                  ▷
Number of cycles required
 Initialize waypoint with home and takeoff location
 **for** 
i=1
 to 
$n_{h}$
 do 
        

$height=h_{min}+\left(i-1\right)\;*\;h_{s}$


 **for** 
i=1
 to 
n
 do

  Add *ABB’A’CDD’C’* location and height to waypoint      ▷ Size of unit
 cycle
  
if

$ceil\left(\frac{ShortEdge}{2s}\right)$
 is odd then    ▷ Odd number means there is still half
 a cycle left to complete
   Add *ABB’A’* location and height to waypoint    ▷ Add the half of
 the last cycle
  **end if**
  Go to the center point of *AA’*   ▷ Do the second cycle at the same
 altitude
  **for** 

i=1
 to 
n
 do
   Add a unit cycle
   **if** 
$ceil\left(\frac{ShortEdge}{2s}\right)$
 is odd then
     Add a half of the last unit cycle
    **end if**
   **end for**
  **end for**
 **end for**
 Transfer waypoint to global coordinates
 Print to readable waypoint file

#### 2.1.3. Circling-Forward Path Planning

As the cycle-Boustrophedon conducts the coverage path utilizing a relatively small track distance, there were some overshoots during the flight due to air turbulence or high airspeed. Therefore, in addition to the cycle-Boustrophedon path, a circling-forward path was introduced into the proposed 3D air quality profiling. For the circling-forward paths, in particular, with a fixed sampling density, a larger distance between two successive parallel tracks was induced, as shown in Figure 11. Therefore, after each altitude level, the final path ended in a ‘circling’ fashion (Figure 12), in which the turnings were all conducted by following an RSR Dubins path with a larger scale than the minimum turning radius.

Moreover, for the waypoints at each iteration, as the distances between them were on a relatively large scale, the effect induced by the waypoint radius could be neglected in this case, whilst the Dubins path could be simply defined by the corner waypoints; this means the final waypoints were the following:
(12)
$$\left\{\begin{array}{c} A=\left(a_{1},a_{2}\right)\\ B=\left(a_{1}-r_{1},\;a_{2}\right)\\ C=\left(a_{1}-r_{1},\;a_{2}+0.5ShortEdge+d\right)\\ D=\left(a_{1}+LongEdge+r_{1},\;a_{2}+\frac{1}{2}ShortEdge+d\right)\\ E=\left(a_{1}+LongEdge+r_{1},\;a_{2}+d\right) \end{array}\right.$$

where point *A* is the starting position of each circling iteration, and ‘ShortEdge’ and ‘LongEdge’, respectively, stand for the shorter edge and longer edge of the ROI rectangle.

The following pseudo-code shows the overall circling-forward path planning Algorithm 2:
**Algorithm** 2 Circling-forward Path Planning.
**Input:**
 Home and takeoff location
 ROI vertices latitude and longitude
 Path separation distance in meters: 
s

 ROI minimum altitude in meters: 
hmin

 Number of layers: 
nh

 Layer separation distance in meters: 
hs

**Output:** Readable waypoint file for GCS software

 Transfer vertices to local coordinates.
 Calculate LongEdge and ShortEdge
 
$n=ceil\left(\frac{ShortEdge}{2s}\right)$
      ▷ Number of cycles required
 **if** 
n
 is odd **then**     ▷ Odd 
n
 results in an uncovered path at middle of ROI
  
n=n+1
       ▷ Round

n
 up to even number
 **end if**
 
$BC=s\;*\;\left(n+1\right)$
          ▷ Size of unit cycle
 
DE=s * n

 
$CD=CD+2r_{1}$
        ▷ Expand LongEdge for full coverage
 Initialize waypoint with home and takeoff location
  **for** 
i=1
 to 
$n_{h}$
 do
    

$height=h_{min}+\left(i-1\right)\;*\;h_{s}$


  **for** 
i=1
 to 
n
 do

  Add *BCDE* location and height to waypoint
  Move unit cycle along *BC* by 
s

 **end for**

**end for**

Transfer waypoint to global coordinates
Print to readable waypoint file

### 2.2. Secure Return to Launch Path Generation

After the coverage path planning, it is considered that the landing path is also essential to ensure the aircraft chooses a safe and proper route. The two key factors of this return to launch path are the controlled descent rate and turn rate. This study proposes an alternate approach for a landing strategy, in which the aircraft circles along the edge of the ROI with a constant and secure descent rate rather than return toward the home position directly. After the UAV descends to a safe altitude, which in our case was 100 above ground, the UAV then flies toward the launch point and switches to the multi-rotor mode to land. The following pseudo-code shows the overall return to launch path generation Algorithm 3:
**Algorithm 3** Return to Launch Path Generation.
 Cycling path completed.
 **while**

height ≥ 100
 **do**
   
$height=height-EdgeLength\;*\;tan\left(\theta \right)\;\;\;\;\;\;\;\;\;\;\;\;\;\;\;\;\;\;\;\;\;\;\;\;\;\;\;\;\;\;\;\;\;\;$
▷ θ is fixed glide angle
   Add next ROI vertex location and height to waypoint
 **end while**
 Add launch location to waypoint
 Transfer to quadrotor mode and land

### 2.3. Easy-Access User Interface

For general utilization, an easy-access user interface was further designed. The user is only requested to input the required ROI for 3D air quality profiling in terms of GPS coordinates while defining the grid dimension, i.e., the profiling density; the application then passes the parameters to the algorithm elaborated in Section 2.1. In addition, the designed module acts as the application programming interface (API) between the proposed algorithm and ground control station software, as it passes the generated waypoints file to the software (‘.waypoints’ file for Mission Planner and ‘.plan’ file for QGroundControl). Figure 13 shows the screenshot of the application’s appearance in which users are allowed to define the vertices (
$v=\left\{v_{1},v_{2},v_{3},v_{4}\right\}$
) of the ROI in the first four entry brackets and the desired altitudes of the ROI in the fifth and sixth entry brackets, as well as the sampling density.

### 2.4. Ground Control Station Software 

Ground control station software usually refers to the software platform for launching and retracting aerial vehicles, which usually acts as a communication module between the pilot and the unmanned aircraft. For most commercial or open-source software, detailed information about a flight mission can be predefined, such as the waypoint coordinates, airspeed at each stage, takeoff and landing methods, and so forth. The software then, either through physical wire or wireless telemetry, sets up the flight commands of the flight controller. As described in Section 2.2, after the user inputs the profiling ROI, the designed module generates a suitable flight path. The flight path then further acts as the waypoints for the ground control station software, and in this study, our application worked well with both QGroundControl and MissionPlanner.

## 3. Simulation Experiment

Both Section 2.1 and Section 2.2 were implemented in Python 3.8.10, in which the U/I was designed with wx-python 4.1.1 library. To validate the proposed coverage path planner, simulations were conducted to compare the auto-grid paths generated by the ground control software and those proposed in this study. In particular, two different simulation platforms were adopted, whose setups are shown in Table 1.

The reason for applying two different platforms was to confirm the applicability of our proposed API with different ground control station software. In short, instead of outputting the flight plan to a physical controller, this experiment was set to run the simulation directly after receiving the waypoint files from the proposed path planner. Then, the result was analyzed in MATLAB after the emulated 3D air quality profiling flight mission. First, the predefined parameters for the simulations are shown in Table 2.

For the rest of this section, as two different platforms were employed, all simulations for the different methods are first presented separately, followed by discussion.

### 3.1. Auto-Grid Paths

#### 3.1.1. Auto-Grid Paths in Simulation Platform 1

By defining the same parameters as presented in Table 2, an auto-grid coverage path was generated in Mission Planner (compared path with Table 1). However, as such functions in most software are mainly designed for photogrammetric purposes, only a 2D coverage path could be acquired. Therefore, after an auto-grid coverage path was generated by Mission Planner (which was based upon the configuration of the FLIR VUE 336 13 mm camera), the path was extended to 3D, where multiple extra layers were added. Additionally, the distance between the tracks was set to the predefined air sampling size, voxel dimension 
d=50 m
. The final generated waypoints are shown in green below in Figure 14.

Figure 15a show the final simulated flight in 2D bird’s eye view. The flight took 4144.7 s to complete the mission, while the total traveling distance was 89, 110 m.

#### 3.1.2. Auto-Grid Paths in Simulation Platform 2

Similar to Mission Planner, an auto-grid coverage path was generated based on the same configuration. The final results show that the simulation flight took 4715.5 s to complete the mission, and the traveling distance was measured to be 94,310 m in total. Below is the bird’s eye view of the simulated flight in Figure 15b.

### 3.2. Cycle-Boustrophedon Path

#### 3.2.1. Cycle-Boustrophedon Path in Simulation Platform 1

By setting all the parameters according to Table 2, the simulation utilizing the proposed cycle-Boustrophedon path planner was carried out. Figure 16 displays the generated waypoints in both 2D and 3D views.

The 2D simulated result is presented in Figure 17a. In this simulation, the flight duration was 5671.8 s, and the flight distance was 110,600 m.

#### 3.2.2. Cycle-Boustrophedon Path in Simulation Platform 2

By applying the same waypoints and the same parameters, the simulation was further conducted in QGroundControl. Similar results were acquired, and Figure 17b shows the final simulation results. The flight duration was 5317.3 s, and the flight distance was 109,300 m.

### 3.3. Circling-Forward Path Planner

#### 3.3.1. Circling-Forward Path Planner in Simulation Platform 1

Applying the same flight parameters shown in Table 2, the proposed circling-forward path planner was simulated under the same scenario. Figure 18 shows the generated waypoints in both 2D and 3D.

The simulation results from Mission Planner are presented in Figure 19a. The total flight duration was calculated to be 6168.2 s, and the flight path distance was 120,095 m.

#### 3.3.2. Circling-Forward Path Planner in Simulation Platform 2

A similar result was acquired by conducting the same mission in QGroundControl, where the flight duration and distance were, respectively, 5802.8 s and 119,280 m. Figure 19b shows the results generated by the QGroundControl simulation platform.

### 3.4. Results and Discussion

From the above, it can be easily observed that with the same user-defined ROI, there existed differences between each method. As seen in Figure 15, the auto-grid path often drifted away from the ROI, especially during the turning points, whereas the proposed methods had better coverage (Figure 17 and Figure 19).

In order to make the discussion of the results more rigorous and credible, a second scenario set was simulated with the same path planners to ensure the repeatability of the simulation results. Table 3 shows the parameters of the two scenarios. The two cubes’ length, width, and height were 600, 800, 300, and 400, 400, 400, respectively.

#### 3.4.1. Coverage Rate of ROI

With two sets of experiments being conducted and to further evaluate the performance of our proposed methods, a few metrics were selected for a more in-depth comparison. It is well known that for a 3D air quality profiling mission, the coverage rate is deemed the most critical aspect of a path design. Therefore, to conduct the numerical appraisement, sampling waypoints were defined. By setting a threshold radius, 15 m in this case, the waypoints could be categorized as “approached” or “bypassed”, based on whether their shortest distances were smaller or equal to or larger than 15 m with the simulated trajectory, respectively. The coverage rate was defined accordingly by:
(13)
$$r_{c}=\frac{N_{A}}{N_{s}}$$

where 
$N_{C}$
 is the number of the ‘approached’ sampling points, while 
$N_{S}$
 is the number of the total sampling points.

Thus, the coverage rate of the two scenarios was calculated, and Table 4 shows the coverage rate of different methods in different simulation software. We can also refer to Figure 20 for visualization, which shows the 3D trajectory with the misapproached sampling points (circled in red). As observed, the missed sampling points were mainly concentrated near the short edges of the ROI, which are the sections where the UAV makes its turns. Thus, it was concluded that our proposed paths, which considered the dynamics of a fixed-wing UAV, improved the trajectory tracking performance. From Table 4, it can be further seen quantitatively that the proposed methods outperformed the auto-grid method in terms of the coverage rate. 

#### 3.4.2. Comparison between Cycle-Boustrophedon and Circling-Forward

Additionally, we can refer to Figure 21 for a closer look at the comparison of the two proposed algorithms. In Figure 21a, which shows the track of the cycle-Boustrophedon method, it can be observed that between every parallel track, the UAV had to rotate its heading angle for 180 degrees, which caused the loose tracking at the short edges of ROI. On the contrary, the circling-forward method, as shown in Figure 21b, had a more considerable distance between every parallel track, in which the heading angle only swung for 90 degrees during every turn. Hence, its tracking performance between every turn outperformed the cycle-Boustrophedon, leading to the best coverage rate amongst all methods in both Mission Planner SITL and QGroundControl-Gazebo simulations. However, it was also deemed that the coverage rate of the cycle-Boustrophedon can be improved by further broadening the size of the short edge extended area. By such a process, not only are the flying distance and duration increased, but it also leads to an expanded area, which results in a lower airspace efficiency (the airspace usage of cycle-Boustrophedon was already larger than cycle-Boustrophedon, which can be observed in Figure 21). On the other hand, the circling-forward method secured a high coverage rate by only containing increased flying distance and duration, which is hence considered the best path planning method in this fixed-wing 3D coverage planning scenario.

#### 3.4.3. Duration and Distance of Different Paths

Engineering analysis for practicality is discussed. To appraise the flying performance in terms of efficiency, the total flying distance and duration are compared in Table 5. Additionally, the discrepancies of the results in terms of percentage are also presented for readers to have a more intuitive understanding of the difference between each method. From the flight duration and distance data, it is obvious that both metrics were significantly larger than the auto-grid method. This trend was caused by the extended flying path out of the ROI from the proposed method. Hence, the extended duration and distance were the trade-offs for coverage rate accuracy. Notably, the long endurance of a fixed-wing VTOL UAVs with a quad-plane configuration should be able to absorb the cost of long flight duration. Furthermore, between the two proposed methods, users can also select different methods between cycle-Boustrophedon and circling-forward based on the trade-off between accuracy and a time requirement.

#### 3.4.4. Return to Launch Paths

Last but not least, Figure 20 also presents the returning routes (colored in orange) of the inspection mission. In Figure 20a,d, it can be observed that different landing strategies were adopted by different firmware. With the ArduPilot firmware in platform 1, the plane flew in fixed-wing mode and spiraled down with a small radius above the home position. In platform 2, which is the PX4 firmware, the fixed-wing UAV transited into the multi-rotor mode and landed above the home position vertically. For a quad-plane configuration UAV, such design of paths could cause instability, whereas it is also propounded that such a UAV application should have a generalized landing method. Therefore, in Figure 20b–f, instead of utilizing the automated returning paths, the proposed return to launch method was integrated and showed similar results. Applying the proposed landing approach, both the descent rate and turning rate were pre-determined and secured, which also prevented divergence between different firmware. This also avoided the confusion of operators while launching UAVs in different firmware platforms and hence reduced operating error and hazard.

## 4. Conclusions

In this study, based on a quad-plane configuration, two path planning methods were studied and proposed to conduct 3D air quality profiling missions, where the objectives were to reach a high coverage rate of a chosen ROI. To achieve this, in particular, the Boustrophedon paths were utilized and extended to 3D, whilst the Dubins paths were also included to ensure the dynamic feasibility. To have a higher coverage rate, the circling-forward planner was proposed additionally to observe the difference in the aforementioned metric. Furthermore, the landing strategy was also included to ensure the robustness of the system; a user interface was embedded with the proposed algorithms, in which the designed module also provided an API to support some of the most popular ground control software platforms. Simulations were then conducted, and by calculating the coverage rate of the sampling points, it was validated that there exists an improvement from the software-generated auto-grid paths to the proposed method’s paths. In addition, the comparison between the two proposed methods was also included, and the duration and distance of all methods were discussed.

With the emphasis on the engineering practicality of this research, it is considered that this study may potentially benefit various parties in both the academic community as well as governmental institutions. Specifically, for the former, researchers in the field of air quality and pollution can harness such a system to collect 3D data to perform profiling and modeling tasks, which were not able to be achieved by conventional methods at such low cost and convenience. Researchers can then use the collected data to conduct further data analysis or learning-based predictions. As for the latter, authorities can have deeper and more thorough monitoring of the environmental condition in both urban and suburban areas; thus, strategic policies can be made, while situational awareness of the climate trend can be grasped. We believe that such a system can not only bridge the gap between theoretical methods and real-world applications but also, to some extent, help with the development of automated environmental protection.

In the near future, a modularized air quality sensor will be equipped on the proposed UAV, i.e., a VTOL quad-plane, while the developed planner will be utilized for the generation of paths. Additionally, a GPS-based synchronization module will also be added so that a complete air quality profile can be established in an efficient fashion. In addition, for extended work, optimization can be considered, where the path planner can be further advanced by minimizing the power consumption in terms of acceleration. Sensor allocation can also be discussed, in which sensitivity analysis between the measurement accuracies and airflow speed can be conducted.

## Figures and Tables

**Figure 1 sensors-22-03630-f001:**
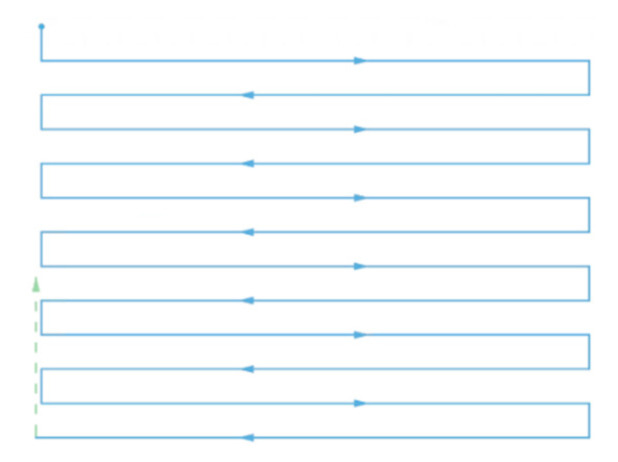
An instance of a Boustrophedon path.

**Figure 2 sensors-22-03630-f002:**
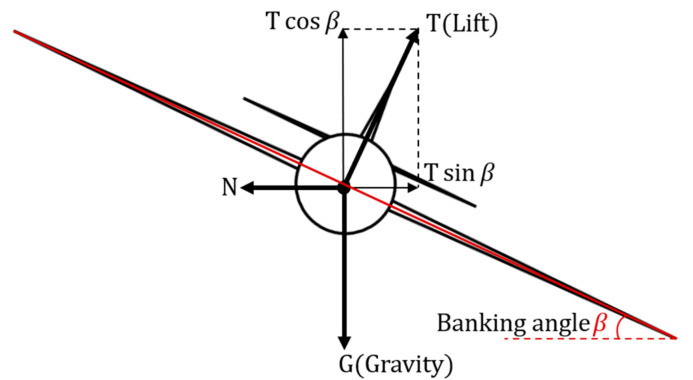
The force diagram of a turning plane with a banking angle of 
β
.

**Figure 3 sensors-22-03630-f003:**
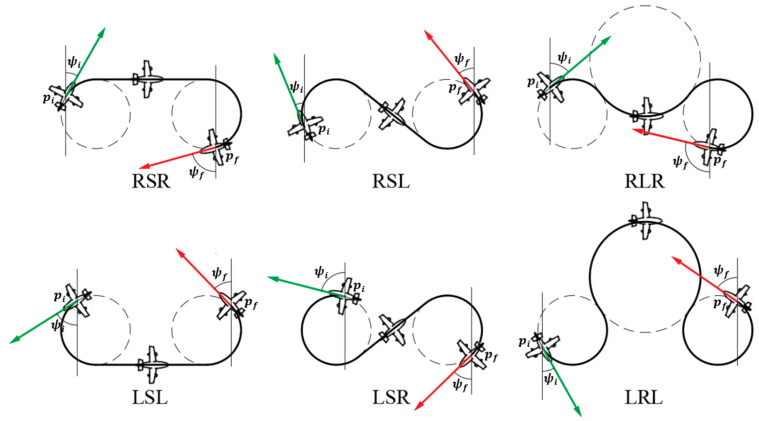
Classification of Dubins paths.

**Figure 4 sensors-22-03630-f004:**
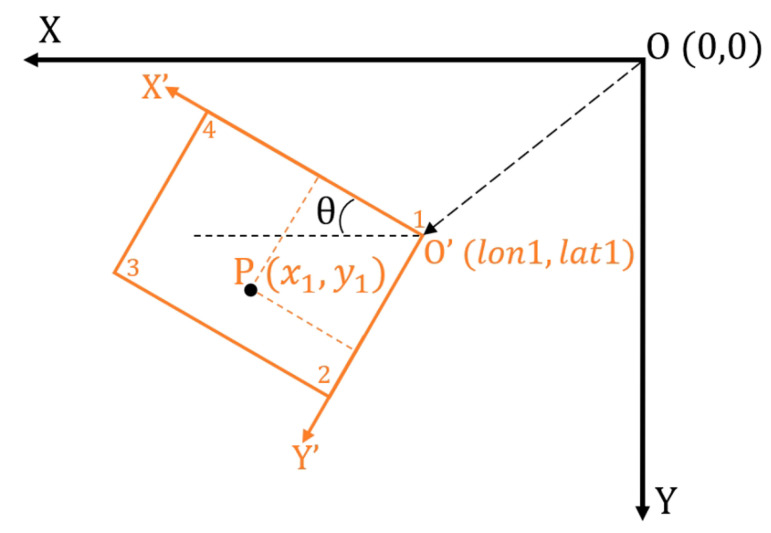
Coordination transformation from GPS coordinates frame to local frame; in which, the *x* axis and the *y* axis of the local frame are defined leftward and downward, respectively.

**Figure 5 sensors-22-03630-f005:**
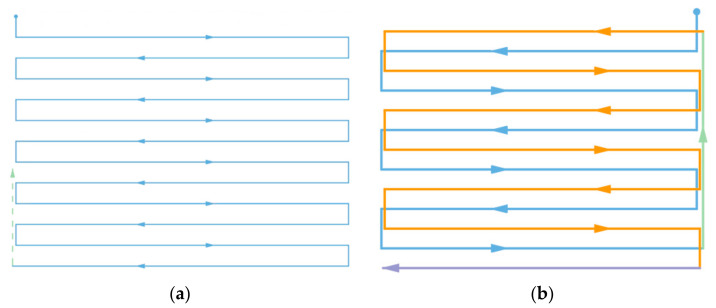
(**a**) Conventional Boustrophedon path and (**b**) the proposed cycle-Boustrophedon path.

**Figure 6 sensors-22-03630-f006:**
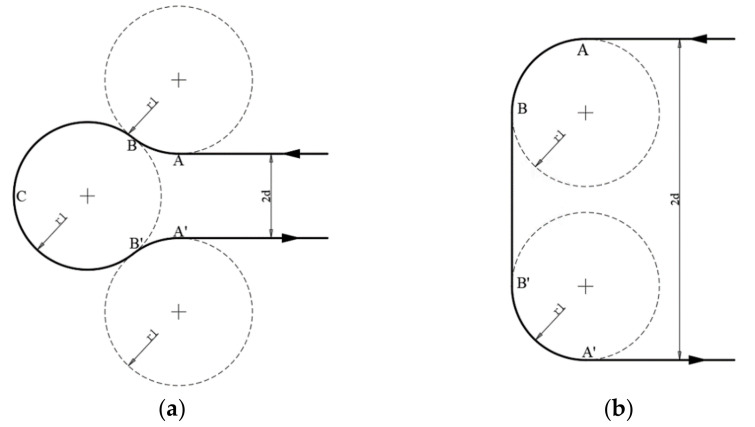
(**a**) illustrates the Dubins path (an RLR instance) when 
d<R
, whereas (**b**) shows the Dubins path (an LSL instance) when 
d≥R
.

**Figure 7 sensors-22-03630-f007:**
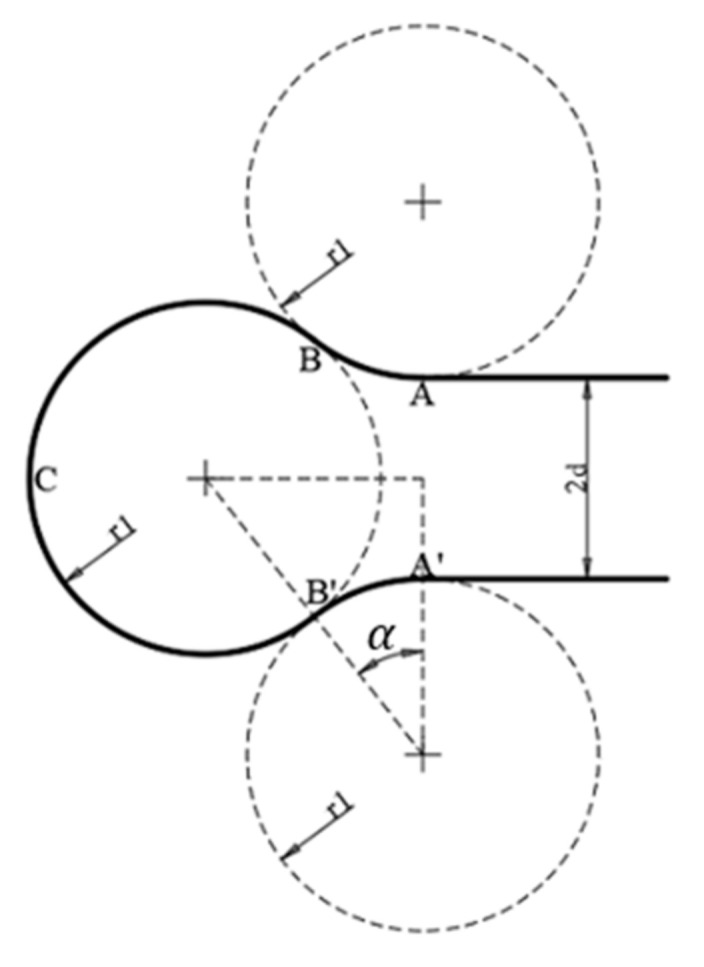
Dubins path (RLR) when 
d<R
, where the aircraft first turns right, then left, and lastly right to complete the Dubins path.

**Figure 8 sensors-22-03630-f008:**
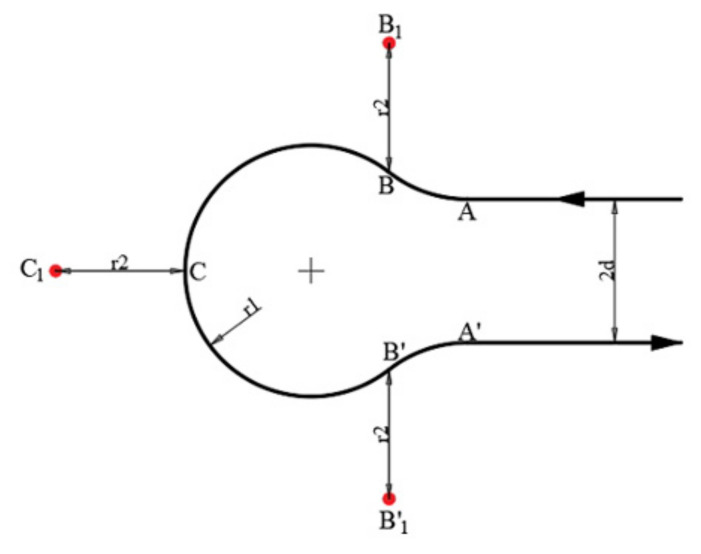
Dubins path (RLR) with the waypoint radius taken into consideration, which is deemed to increase the turning performance, resulting in a better trajectory.

**Figure 9 sensors-22-03630-f009:**
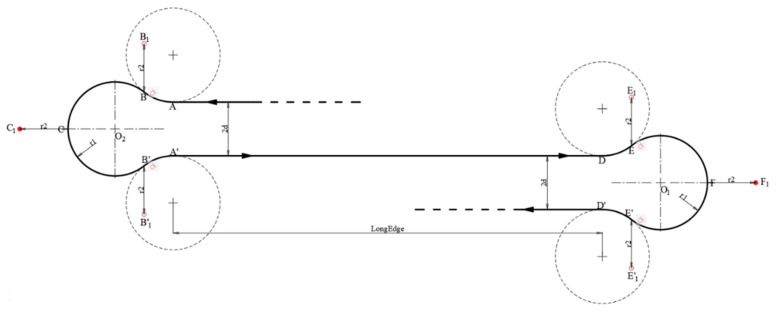
Dubins path (RLR + LRL) for each iteration.

**Figure 10 sensors-22-03630-f010:**
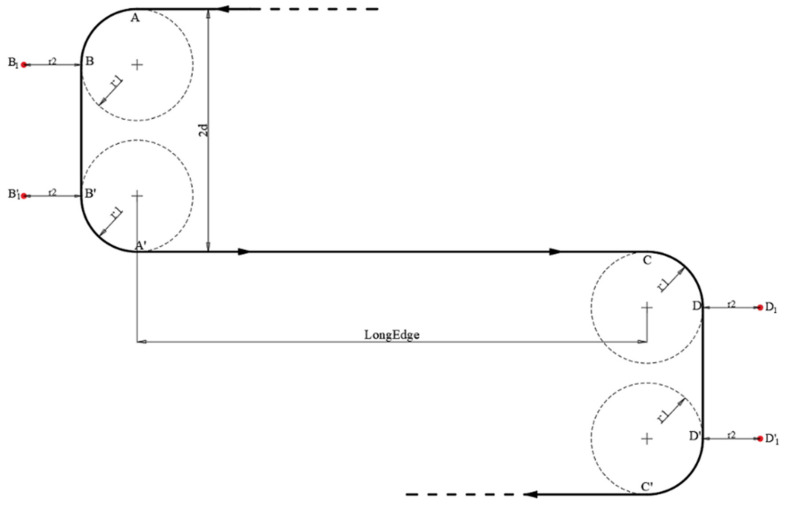
Dubins path (LSL + RSR) when 
d≥R
 for one iteration.

**Figure 11 sensors-22-03630-f011:**
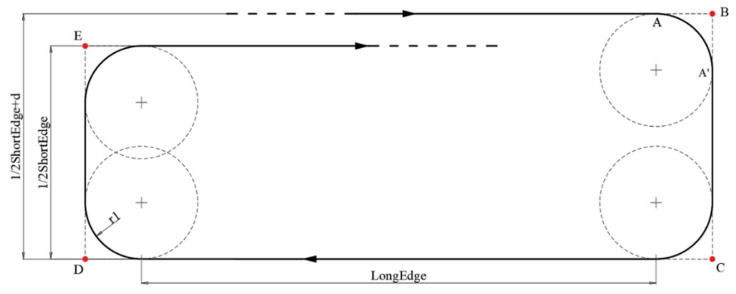
One iteration of the circling-forward path.

**Figure 12 sensors-22-03630-f012:**
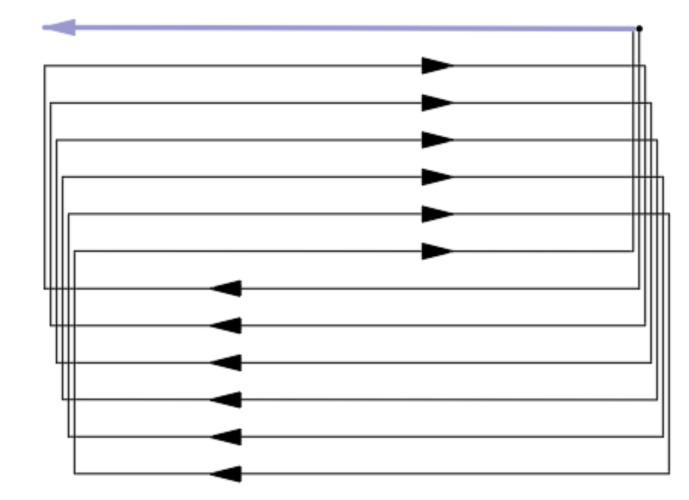
The final circling-forward path.

**Figure 13 sensors-22-03630-f013:**
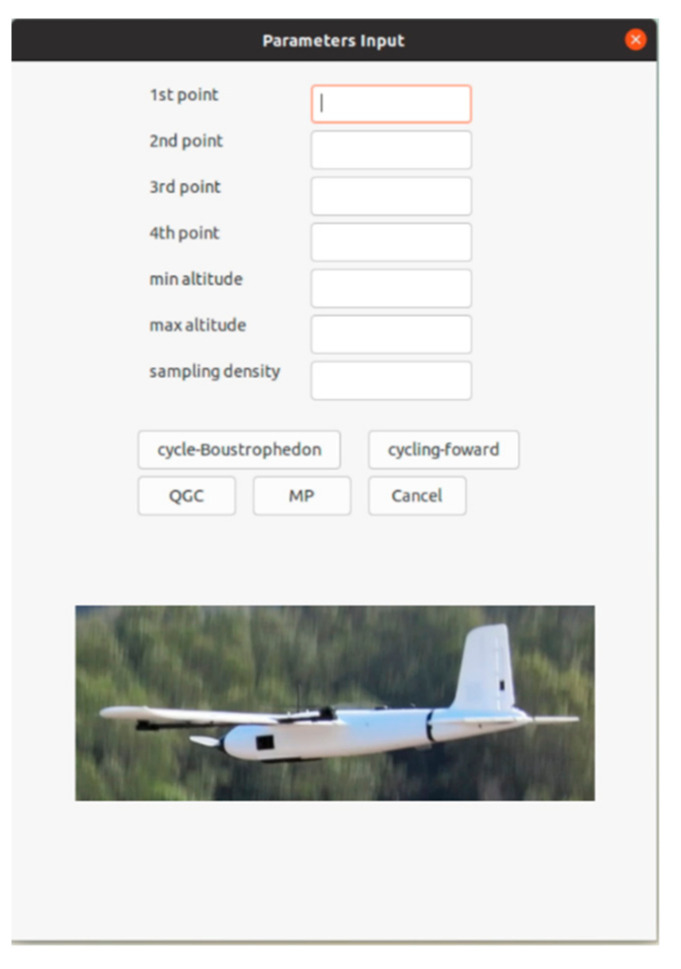
The designed GUI for air quality profiling coverage path planning.

**Figure 14 sensors-22-03630-f014:**
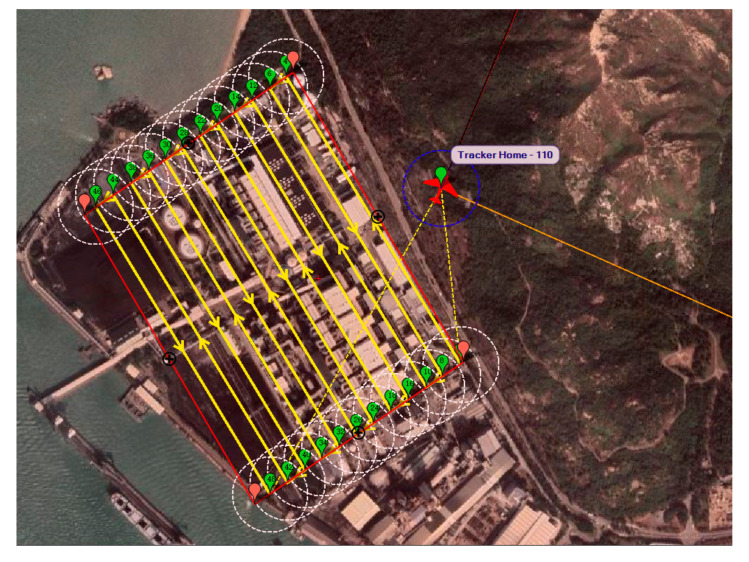
Auto-grid coverage paths generated by Mission Planner.

**Figure 15 sensors-22-03630-f015:**
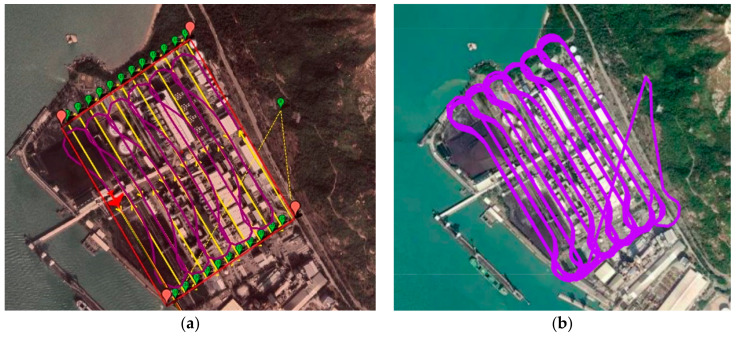
Bird’s eye view of the trajectory result of auto-grid paths in simulation platform 1 (**a**) and platform 2 (**b**).

**Figure 16 sensors-22-03630-f016:**
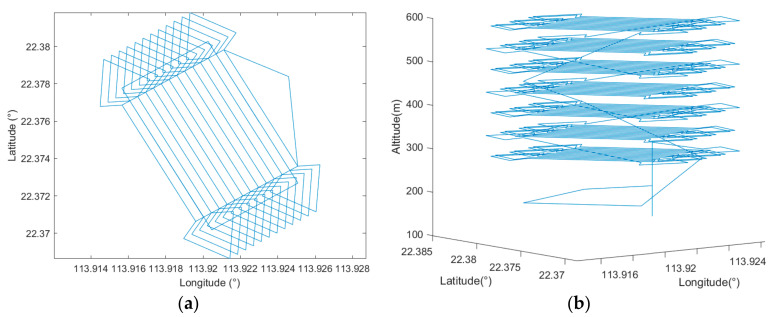
(**a**) Bird’s eye view of the cycle-Boustrophedon waypoints and (**b**) the 3D visualization.

**Figure 17 sensors-22-03630-f017:**
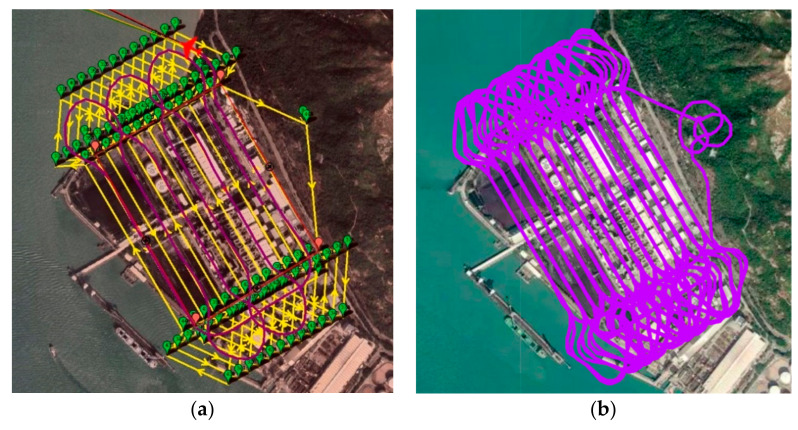
Bird’s eye view of the final trajectory result of cycle-Boustrophedon path in simulation platform 1 (**a**) and platform 2 (**b**).

**Figure 18 sensors-22-03630-f018:**
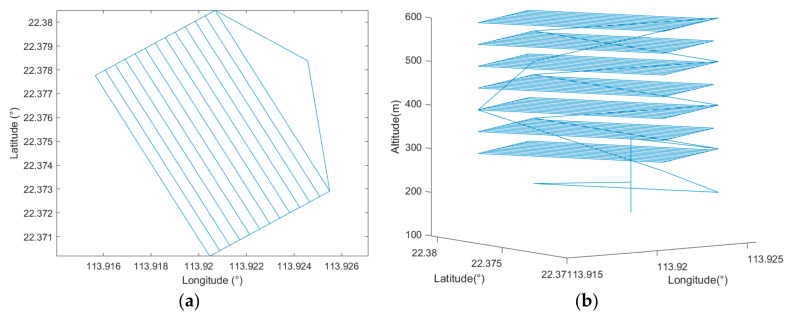
(**a**) Bird’s eye view of the circling-forward waypoints and (**b**) the 3D visualization.

**Figure 19 sensors-22-03630-f019:**
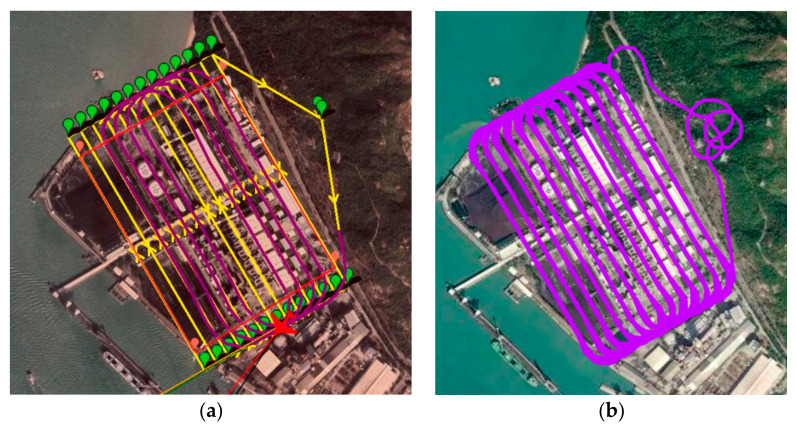
Bird’s eye view of the final trajectory result of circling-forward path in simulation platform 1 (**a**) and platform 2 (**b**).

**Figure 20 sensors-22-03630-f020:**
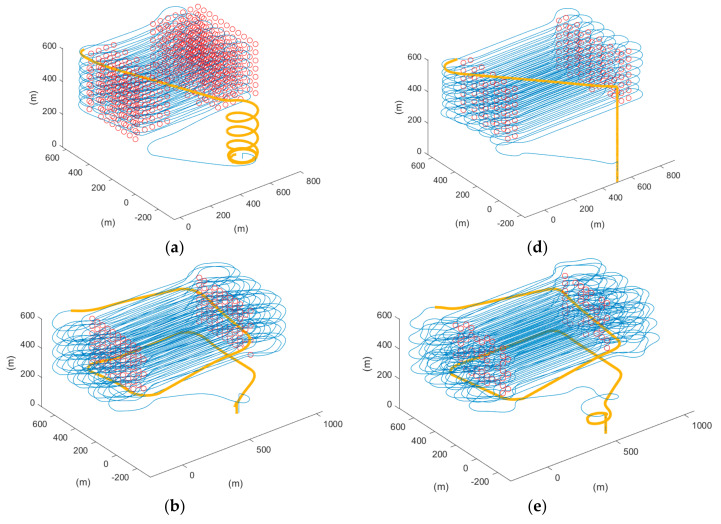
The 3D trajectory results of scenario 1 in blue lines, the not approached sampling points in red circles, and returning route in orange bold lines of (**a**) auto-grid paths, (**b**) cycle-Boustrophedon, and (**c**) circling-forward in simulation platform 1, and (**d**) auto-grid paths, (**e**) cycle-Boustrophedon, and (**f**) circling-forward in simulation platform 2.

**Figure 21 sensors-22-03630-f021:**
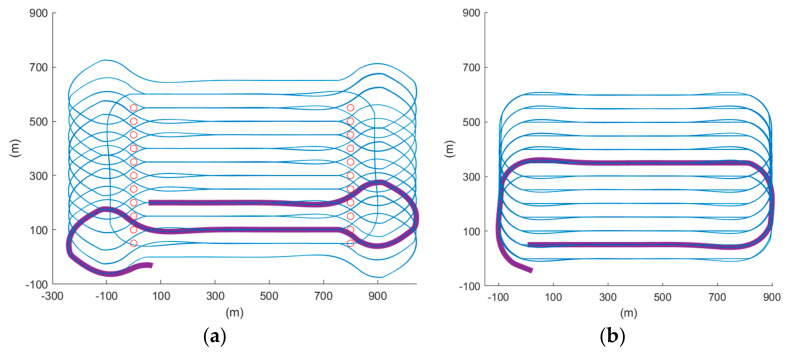
The 2D trajectory results of scenario 1 in blue lines, the not approached sampling points in red circles, and the turning route between first two parallel tracks in bold lines of (**a**) cycle-Boustrophedon (**b**) circling-forward in simulation platform 1.

**Table 1 sensors-22-03630-t001:** Simulation setup.

	Simulation Platform 1	Simulation Platform 2
Software Platform	Mission Planner SITL (software in the loop) Simulation [34]	Gazebo [40]
Firmware	ArduPilot	PX4
Compared Paths	Auto-grid paths by Mission Planner	Auto-grid paths by QGroundControl

**Table 2 sensors-22-03630-t002:** Predefined parameters.

Parameters	Value
$V_{max}$ (max. air speed)	30 m/s
$V_{min}$ (min. air speed)	10 m/s
β (banking angle)	25 °
$r_{1}$ (turning radius, assumed)	87.5 m
$r_{2}$ (waypoint radius)	90 m
d (sampling density)	50 m
$v_{1}$ (vertex 1)	(113.9250, 22.3736)
$v_{2}$ (vertex 2)	(113.9202, 22.3705)
$v_{3}$ (vertex 3)	(113.9163, 22.3768)
$v_{4}$ (vertex 4)	(113.9211, 22.3798)
$h_{max}$ (max. altitude)	600 m
$h_{min}$ (min. altitude)	300 m

**Table 3 sensors-22-03630-t003:** Parameters of the simulation scenarios.

	Scenario 1	Scenario 2
Vertex 1 coordinates	(113.9250, 22.3736)	(114.2672861, 22.34372)
Vertex 2 coordinates	(113.9202, 22.3705)	(114.2711671, 22.3439472)
Vertex 3 coordinates	(113.9163, 22.3768)	(114.2713736, 22.3403549)
Vertex 4 coordinates	(113.9211, 22.3798)	(114.2674926, 22.3401327)
$h_{max}$	600	500
$h_{min}$	300	100

**Table 4 sensors-22-03630-t004:** Comparison of the coverage rate between different methods on different platforms.

	Scenario 1		Scenario 2	
	MP SITL	QGC-Gazebo	MP SITL	QGC-Gazebo
Auto-grid	53.78%	90.99%	50.97%	71.43%
Cycle-Boustrophedon	89.69%	93.89%	83.07%	87.65%
Circling-forward	100%	100%	100%	100%

**Table 5 sensors-22-03630-t005:** Comparison of the flight duration and distance between different methods on different platforms.

	Scenario 1							
	MP SITL				QGC-Gazebo			
	Flight duration		Flight distance		Flight duration		Flight distance	
Auto-grid	4145 (s)		89.11 (km)		4716 (s)		94.31 (km)	
Cycle-Boustrophedon	5672 (s)	+36.84%	110.6 (km)	+24.11%	5317 (s)	+12.76%	109.3 (km)	15.89%
Circling-forward	6168 (s)	+8.75%	120.1 (km)	+8.58%	5803 (s)	+9.13%	119.3 (km)	9.13%
	**Scenario 2**							
	**MP SITL**				**QGC-Gazebo**			
	Flight duration		Flight distance		Flight duration		Flight distance	
Auto-grid	2291 (s)		38.44 (km)		2842 (s)		48.30 (km)	
Cycle-Boustrophedon	3527 (s)	+53.95%	54.82 (km)	+42.61%	3077 (s)	+8.27%	57.73 (km)	+19.52%
Circling-forward	3768 (s)	+6.84%	59.57 (km)	+8.67%	3342 (s)	+8.61%	62.85 (km)	+8.87%

## Data Availability

Not applicable.

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
