# Peer review of "Development of Fixed-Wing UAV 3D Coverage Paths for Urban Air Quality Profiling"

_sensors, 2022, doi:10.3390/s22103630_

Round 1

Reviewer 1 Report

The paper proposes a 3D coverage path planning for unmanned aerial vehicles and designs a human-machine interface. The research is innovative and potentially adds useful knowledge in the field of UAVs. However, there are some major concerns that need to be addressed before being considered for publication.

Introduction, first paragraph, the Global air pollution issue should be also mentioned before discussing the regional issue.

The literature review regarding conventional air monitoring methods, e.g., the fixed location and ground-based is very pool. More in-depth review should be added to discuss the pros and cons of these methods. Relevant citations should be added. The review should be not only limited to the monitoring range but also cover a wide range of aspects for these methods, for example, the cost, the accuracy of monitoring, the maintenance, the data quality (accuracy, timeliness and accessibility).

Again, the literature review regarding existing UAV methods also need significant improvement. What are the key strength and limitations of each study cited? This is important to shape the research gap of the study.

The paragraph about the software (line 76 to 85) should be moved to methods.

Where is the review of the literature regarding the user interface design of the UAV? Why this is necessary? This should be covered in the literature review since it is one important objective of the paper.

The results section is very detailed and clear.  Has the userface been evaluated in any way?

However, the discussion section is quite poor. It needs rewrite. It reads like Results, NOT Discussion. The discussion section should talk about how to interpret the results. How are results compared with existing/previous studies of UAVs? How do the findings of this research stand among the existing literature?

The conclusion also needs to be improved. What are the implications of the current study for policymakers, manufacturers, and academia? What is the limitation of this study?

Author Response

Please find the attached file for authors' reply. 

Reviewer 2 Report

This paper focuses on the development of 3D coverage path planning based upon current commercial ground control software, where the method mainly depends on the Boustrophedon and Dubins paths. The work is very interesting and novel. I have some following questions:

  • The novelty of the paper should be outlined one by one.
  • Line 110, the authors get three advantages. However, there are no references to support the idea.
  • The authors developed the designed GUI. The author should list the program platform.
  • In Figure 6, two different path methods are compared. In the experiment sections, the two methods should be compared too. So that, the advantage of the proposed method cycle-Boustrophedon can be further improved.
  • In line 264, the authors develop GUI based on the proposed algorithm. You’d better list the detail of the proposed algorithm.
  • The introduction of the paper should be improved.

Reviewer 3 Report

Various parts of the manuscript needs to be reconsidered:

  • I would recommend to slightly reconsider the keywords of the manuscript. More and shorter keywords (i.e. not keywords, which consists of 4-5 words) would be better.
  • I would recommend to extend the literature review. Higher number of scientific research papers should be analysed.
  • Regarding the structure of the paper, in section "Introduction" figure 1 looks excessive and unnecessary. I would recommend to remove this figure.
  • Major drawback of the manuscript – analysis and discussions part. Both of these part are too limited. More deep scientific analysis should be performed. 
  • Conclusions also needs to be reconsidered. As it is now, conclusions do not properly represents the performed research. 

Round 2

Reviewer 1 Report

The authors have addressed my concerns and the manuscript has been improved. I have no further comments. 

Author Response

We would like to show our most sincere gratitude to the editor's invaluable comment and effort.

Reviewer 2 Report

The paper is well revised. Now, the version is much better than the previous one. The introduction can be further improved. Some references miss, such as 

https://doi.org/10.1016/j.asoc.2020.106857. 

Author Response

We thank the suggestion from the reviewer, and have modify our manuscript again accordingly by adding the following references,

From line 96 to 102 of the revised manuscript:

In addition to all the above, Yu et al. [32] utilized the knee-guided differential evolution algorithm [33] to construct a path planning problem for UAVs for disaster scenarios within 3D terrain situations. In particular, the work aimed to solve the problem by utilizing B-Spline paths and modeling the task with multi-objective optimization, where distance and risk are mainly considered as the objective functions; such a method could then offer an optimal solution and allow the vehicle to follow a smooth path.

We hope that this could make the introduction section of our manuscript complete, and further reach the standard of this journal publication. 

Reviewer 3 Report

No new comments. Good luck with Your future research. 

Author Response

We would like to show our most sincere gratitude to the invaluable comments from the reviewer.